# A Phylogenetic and Ontogenetic Perspective of the Unique Accumulation of Arterial Variations in One Human Anatomic Specimen

**DOI:** 10.3390/medicina56090449

**Published:** 2020-09-04

**Authors:** Bettina Pretterklieber, Michael L. Pretterklieber

**Affiliations:** Division of Anatomy, Center for Anatomy and Cell Biology, Medical University of Vienna, A-1090 Vienna, Austria; michael.pretterklieber@gmail.com

**Keywords:** anatomic variation, arteries, phylogeny, ontogeny

## Abstract

*Background and objectives:* Anatomical dissection is an indispensable means of acquiring knowledge about the variability of the human body. We detected the co-existence of several arterial variations within one female anatomic specimen during routine anatomical dissection. The aim of this study was to evaluate if this status is a regular pattern in any of other vertebrates. *Materials and Methods*: Besides of a meticulous anatomic dissection, we performed a literature review concerning the frequency, the phylogenesis, and ontogenesis of all of these variations. *Results*: Exceptionally, the middle colic artery arose from an extraordinarily divided celiac trunk. The kidneys received three polar arteries. On the left side, a corona mortis replaced the obturator artery. The aortic arch gave rise to a bicarotid trunk, and the right subclavian artery originated and coursed as a typical lusorial artery leading to a non-recurrent laryngeal nerve on the right side. Furthermore, variations of the branches of the thyrocervical trunk were found to be present. Extraordinarily, in their cervical portion both internal carotid arteries gave rise to two arteries each. All of these variations developed within two to three weeks, around the sixth week of gestation. It was not possible to ascribe all or even one of the variations to a singular species of vertebrates. *Conclusion*: Apparently, arterial variations are frequently a result of random development. Medical professionals must always be aware of anatomical variations; the absence of such awareness would create major difficulties during surgery. The present case confirms the relevance of anatomical dissection, particularly for medical students.

## 1. Introduction

As each human body is unique in terms of structure, anatomical variations are rather the reality than the “Vesalian” view described in anatomy textbooks [1]. Unfortunately, medical students are mainly required to learn about the “normal” pattern, which means that future physicians may think muscles, vessels, nerves, and bones are arranged almost identically in all human beings. As stated by Bergman [1], much anatomical information—especially from the previous centuries—is ignored or lost and is then constantly “rediscovered” in studies by authors who often do poor library searches. The same “rare” anatomic variations encountered in anatomical dissection, radiography, or surgery are then described repeatedly in case reports. A detailed search of books and monographs written by anatomists (such as Gruber, Haller, Henle, Hyrtl, LeDouble, Testut, Wood, and others) would reveal that the large majority of these “new” anatomic variants have been described in the 19th century [1]. Nevertheless, it would be appropriate to publish cases of rare anatomical variations because, in former times, the authors illustrated their descriptions at best with line drawings. Our contemporary mode of photographic documentation is a vivid means of sharing such findings with other anatomists and physicians, thus ensuring their awareness of these conditions in their clinical routine.

Bergman et al. [2] have given us a large encyclopedia of anatomic variations. This work serves as a very important source of preliminary knowledge about the frequency or names of (rare) variations. We have observed and documented well-known variations during our student dissection courses, such as a corona mortis or a sternal muscle. Occasionally, we encounter rare variants worthy of being published with detailed photographs. Furthermore, for an anatomist, the potential phylogenetic and ontogenetic background of these variations is interesting to know.

Here we present the interesting condition within one female anatomic specimen, which was dissected in our gross anatomy course. This individual had not just one, but a large number of variations and even rare patterns of the gastrointestinal arteries, arterial blood supply to the kidneys, the obturator artery, branches of the aortic arch, and branches of the subclavian and external carotid arteries. The aim of this study was to elicit the frequency of these anatomical variations by an in-depth literature review. Furthermore, we wanted to clarify, if such an arterial pattern is the regular situation within another species of vertebrates. Therefore, brief accounts of comparative anatomy are provided to explain the phylogeny of these anatomical structures. In order to determine whether these variations had been established at the same point of development probably caused by an external influence, we reviewed the embryological formation of these structures.

## 2. Materials and Methods

During our student dissection course, we observed several anatomical arterial variations in the specimen of a 97-year-old Caucasian woman. She died on myocardiopathy, pneumonia, and sepsis. We lack any further clinical information. The specimen has been fixed with a mixture of 5% phenol and 1.6% formalin via arterial perfusion and subsequent immersion. Within the regions, in which arterial variations were present, we did not observe any other anatomical irregularities. Furthermore, no signs of surgical intervention were noted at these sites. The deceased woman had donated her body to medical education and research at our institution. In addition to the informed consent of the deceased individual, approval was obtained from the ethics committee of our university (approval number: 1813/2017 dated 20 October 2017).

Due to the schedule of our dissection course, the arterial supply of the intraperitoneal organs was dissected in situ. Thereby all accompanying veins were carefully removed. Then, the intraperitoneal together with the secondary retroperitoneal organs were removed en bloc in connection with their mesos. Thus, we got access to the retroperitoneal space. After dissecting the regions of the neck and head until the parapharyngeal space, the rostral part of the head (the face and the base of the skull anterior to the foramen magnum) in connection with the organs and all neurovascular bundles anterior to the vertebral column were mobilized. The latter were transected at the level of the superior thoracic aperture. Finally, the retropharyngeal and parapharyngeal spaces, the pharynx, and the larynx were dissected. After recognizing the exceptional arterial variation concerning the celiac trunk and superior mesenteric artery, we paid special attention during the ongoing dissection.

After careful dissection, we took photographs with a digital reflex camera (Canon EOS 5D Mark II, Canon Inc., Tokyo, Japan). Cropping and labeling of the pictures were performed with a standard professional image editing program (Adobe Photoshop CS6 extended Version 13.0; Adobe Systems Inc., San José, CA, USA).

Using the database of Bergman et al. [2], relevant monographs, and original articles concerning arterial variations in humans, we clarified the frequency of occurrence of all arterial variations observed. We considered the embryogenesis with special emphasis on the question if all these variations have been developed at the same stage, maybe influenced by an external factor. Furthermore, we looked whether the combination of these variations is a common pattern in other vertebrates.

## 3. Results

### 3.1. Replaced Middle Colic Artery and Split Celiac Trunk

During dissection of the superior mesenteric artery, we found the jejunal, ileal, ileocolic, and right colic arteries in regular arrangement. Although we observed a reasonably strong artery in the transverse mesocolon, equivalent to a middle colic artery, it lacked any connection with the superior mesenteric artery or one of its branches at the regular site of its origin, i.e., the ventral aspect of the superior mesenteric artery. Following this vessel to its origin, we found that it disappeared into the pancreatic notch (Figure 1a). As we exposed the celiac trunk and its branches, we found the origin of the replaced middle colic artery together with the splenic artery from an unusually formed celiac trunk (Figure 1b). Within the pancreatic notch, the replaced middle colic artery—from its left side—gave rise to the dorsal pancreatic artery. The latter divided into a smaller right branch and a larger left branch. The right branch was connected to the prepancreatic artery, which in turn anastomosed with the right aspect of the middle colic artery approximately at the level of origin of the dorsal pancreatic artery. The left branch constituted the inferior pancreatic artery. The peripheral aspect of the middle colic artery coursed in regular fashion towards the right colon flexure and revealed a regular branching pattern. Thus, it formed a right branch to anastomose with the ascending branch of the right colic artery, and a left branch to connect with the ascending branch of the left colic artery.

The celiac trunk was separated into two trunks originating independently from the ventral aspect of the abdominal aorta, just below the aortic hiatus (Figure 2). The first branch of its left and stronger part was the left inferior phrenic artery. Then the splenic and replaced middle colic arteries arose in a trouser-shape formation. Finally, it continued into the common hepatic artery. The latter gave rise to the right gastric and gastroduodenal arteries, and terminated into the hepatic artery proper which divided into three branches. The right branch supplied the right lobe, the left branch the main part of the left lobe, and the intermediate branch entered the quadrate lobe of the liver. The cystic artery originated from the right branch. The smaller right portion of the celiac trunk subdivided into the left gastric and the right inferior phrenic arteries. From the left gastric artery, an aberrant left hepatic artery arose and ran through the dense part of the hepatogastric ligament to supply a small part of the left lobe of the liver. The left gastric artery reached the lesser curvature of the stomach via the gastropancreatic fold.

### 3.2. Renal Polar Arteries

Both kidneys received supernumerary arteries (Figure 3). On the left side, the additional renal artery originated from the lateral aspect of the abdominal aorta, approximately 1.5 cm above the origin of the inferior mesenteric artery. It coursed dorsally to the left ovarian vein and the inferior branch of the left renal vein, and ventrally to the left ureter. Finally, it divided into an upper branch entering the hilum supplying the anterior inferior segment, and a lower branch constituting a polar artery for the inferior segment. Three arteries supplied the right kidney: one of the additional arteries originated from the ventral aspect of the abdominal aorta, just above the regular renal artery, at the level of the superior mesenteric artery. It gave rise to an upper polar artery supplying the superior segment. The two lower branches entered the hilum and supplied the two anterior segments. A lower polar artery for the inferior segment originated from the ventral aspect of the abdominal aorta, at the level of the inferior mesenteric artery. It coursed between the inferior vena cava and the right ovarian vein, and dorsally to the right ureter.

### 3.3. Corona Mortis

An enlarged pubic branch from the left inferior epigastric artery, forming a so-called corona mortis, replaced the left obturator artery. It then left the lesser pelvis in a regular manner through the obturator canal.

### 3.4. Aberrant Right Subclavian Artery and Bicarotid Trunk

The branches from the convex aspect of the aortic arch originated as follows (Figure 4a,b). The first was a common trunk of both common carotid arteries, the second the left subclavian artery, and the last—coming from the dorsal aspect of the arch—the right subclavian artery following a retro-esophageal course to reach the right posterior scalene space. Consequently, the right recurrent laryngeal nerve was absent. The right inferior laryngeal nerve branched off directly from the right vagus nerve and reached its site of entry in the cricopharyngeal portion of the inferior pharyngeal constrictor, coursing horizontally. The thoracic duct divided within the range of its terminal bend and united again before it ended in the left venous angle (Figure 4c). The right vertebral artery originated a little more medially than usual and coursed in an S-shaped manner to reach the foramen transversarium of the sixth cervical vertebra (Figure 4a,b).

### 3.5. Variations in the Formation of the Thyrocervical Trunk

The right thyrocervical trunk gave rise to the internal thoracic artery. Then it subdivided into the inferior thyroid and transverse cervical arteries. The ascending cervical artery originated from the latter (Figure 5a). The right suprascapular artery arose independently from the third part of the subclavian artery. On its course, a root of the lateral pectoral nerve and the anterior division of the superior trunk of the brachial plexus crossed superficially. The suprascapular artery then coursed below the superior transverse scapular ligament to enter the supraspinous fossa (Figure 5b).

### 3.6. Aberrant Branches of the External Carotid Artery

The cervical portion of both internal carotid arteries had exceptional branches. The right internal carotid artery gave rise to the right occipital and the ascending pharyngeal artery (Figure 6a), and the left one to a common stem of the ipsilateral ascending pharyngeal and ascending palatine arteries (Figure 6b).

### 3.7. Ontogenesis and Phylogenesis

The timetable of development of all arterial variations observed in this study is shown in Table 1. As the definitive arterial pattern develops around the sixth week of gestation, the critical period for the formation of the observed variations lasts between two and three weeks, i.e., the fifth to eighth week.

Table 2 summarizes in which species the observed variations have been reported to be the regular pattern. It is not possible to ascribe them to a specific species of vertebrates. Further information concerning the ontogenetic and phylogenetic aspects is given in the discussion.

## 4. Discussion

### 4.1. Replaced Middle Colic Artery and Split Celiac Trunk

As a rule, the middle colic artery is a branch of the superior mesenteric artery, arising from its ventral aspect below the pancreatic notch [4,5]. In certain cases—as in the present one—the middle colic artery may arise from the celiac trunk or one of its branches [3,5,6,7,8,9,10,11]. It has also been found to originate from the inferior mesenteric artery [12] or directly from the aorta [13,14]. Several authors described these variants as single observations during anatomical dissection. It arose, in addition to the regular three branches from the celiac trunk [11], from a “pentapus” (celiac trunk with five instead of three branches) [8], from the proximal segment of the splenic artery [6] or the proximal segment of the common hepatic artery [7,10]. One author detected a replaced middle colic artery arising from the celiac trunk during a celiac angiography performed in a 12-year-old girl [9]. Nevertheless, a middle colic artery arising from the celiac trunk is a rare condition in humans. Michels [5] found it in 1% of 200 cases. In their standard works about arterial variations, Quain [15], Adachi [16], and Lippert and Pabst [4] did not mention this rare condition. A common stem with the splenic artery showing a trouser-shaped formation, as in the present case, has not been reported previously.

A so-called accessory middle colic artery may be present in about 9–18% of individuals, arising directly from the superior mesenteric artery or the regular middle colic artery [2,4,5,16]. This artery supplies the distal third of the transverse colon and the left colic flexure. It divides into two main branches, which unite with the left branch of the middle colic artery and the ascending branch of the left colic artery, respectively. The latter connection may course parallel to the marginal branch (part of the marginal artery of Drummond) of the left colic artery, forming the rare “arch of Riolan” [5]. This accessory middle colic artery may, in exceptional cases, originate from the celiac trunk or a part of it [2,5,17], such as the middle colic artery in our case. Adachi [16] observed this only in one case. 

Haller [18] was the first anatomist to describe the regular pattern of the celiac trunk. This so called “Tripus Halleri” is a trifurcation into the common hepatic, splenic, and left gastric arteries [4,15,16]. However, it is not always an actual trifurcation because the left gastric or the common hepatic artery may arise before the trunk divides into the two remaining vessels [15,16]. A celiac trunk with three branches has been found in 75–90% of individuals [3,4,5,16,19,20]. Variations of the celiac trunk are quite common and have been consistently described over the last centuries [2,8,15,20,21,22]. Thus, several classifications have been made to describe and quantify the different patterns of formation [4,5,16,19]. In 7–25%, one of the three main branches arises separately from the abdominal aorta, resulting in an incomplete celiac trunk [3,4,5,15,16,19,20]. An exceptional situation is the total absence of a trunk formation, with all branches originating directly from the abdominal aorta [15,23,24]. Lippert and Pabst [4] reported a frequency of less than 1%; whereas Adachi [16] and Michels [5] did not mention this rare variation.

Besides the regular anastomosis between the vascular territory of the celiac trunk and the superior mesenteric artery (via the pancreatic and duodenal arteries) [5,16,17] as well as the above-mentioned uncommon condition of a replaced middle colic and accessory middle colic arteries, additional anastomoses may occur. An aberrant right hepatic artery may originate from the superior mesenteric artery in 10–17% [5,16]. Exceptionally, the common hepatic artery is known to arise with the superior mesenteric artery [3,22] in a so-called hepatomesenteric trunk at a rate of 0.4–3% [4,5,16]. Moreover, several authors described a direct “anastomotic channel” between the celiac trunk and the superior mesenteric artery. Like the replaced middle colic artery in the present case, it is situated behind the pancreas and the inferior mesenteric vein, and anastomoses frequently with the middle colic artery near its origin [16,25,26,27] or even with one of the jejunal arteries [17]. Probably, the anlage of this channel may also lead to a replaced middle colic artery as in the present case.

A common origin of the celiac trunk and the superior mesenteric artery may exist [2,15,21,22], and has been reported to occur in 0.4–2.4% of cases [3,4,5,16,20]. Several authors have observed this variation [28,29]. As a rare condition, the superior mesenteric artery may have a common origin with only one or two arteries of the celiac trunk, forming a splenomesenteric trunk in 1%, a hepatomesenteric trunk in 0.4–3%, a hepatosplenomesenteric trunk in 0.5–1%, or a gastromesenteric trunk [4,5,16,20,30]. However, such a separation of the celiac trunk has not been reported when the middle colic artery alone arose from the celiac trunk, as in the present case which revealed a hepatosplenocolic trunk giving rise to the left inferior phrenic artery.

An aberrant left hepatic artery originating from the left gastric artery is a fairly common variation, occurring in 15–30% of cases [4,5,16]. Michels [5] observed in one half of cases, and Adachi [16] in two thirds of cases, a so-called replaced left hepatic artery with an absent regular left hepatic artery, supporting the entire left lobe of the liver. The remaining cases have been so-called accessory left hepatic arteries with a regular—but usually smaller—left hepatic artery [5,16], as in the present case. In addition, the left gastric artery together with the right inferior phrenic artery formed a gastrophrenic trunk, individually arising from the abdominal aorta, which has not been reported previously.

The origin of one or both (either separate or with a common stem) inferior phrenic arteries from the celiac trunk is a common variation in humans and has been observed at a rate of 46–74% [4,5,15,16]. Occasionally (5–9%) an inferior phrenic artery may originate from the left gastric artery as well [4,16]. As in our case with a split celiac trunk, the left inferior phrenic artery arose from the hepatosplenicocolic trunk, whereas the right one formed a gastrophrenic trunk with the left gastric artery, which in turn gave rise to an accessory left hepatic artery.

The present case is unique: the formation of a hepatosplenocolic and a gastrophrenic trunk has not been described previously. In cases with a separated celiac trunk, the additional artery has been the superior mesenteric artery and not a replaced middle colic artery as one of its branches [22]. Moreover, in the hitherto reported cases a replaced middle colic artery always originated from a complete celiac trunk [6,8,9,11].

The vessels supplying the gut originally arise in a metamere manner from the dorsal aorta. The more complex the changes in the intestines, the more frequently the number of arteries is reduced to three arteries as in humans [22,31,32]. Thus, the hagfish, the cartilaginous fish, and the teleost have numerous vessels to the gut, whereas the pattern in the salamander and in amphibians becomes similar to that of higher vertebrates. The so-called celiacomesenteric trunk is a regular occurrence in many saurians (particularly in the lizard), in salientians (such as the frog), some whales, the mole, in monotremes (anteater and platypus), and occasionally in bats. The superior mesenteric artery is already separated from the celiac trunk in the tuatara, in turtles, marsupials, insectivores, carnivores, rodents, ungulates, and primates [22,31,32,33]. Thus, connections like a replaced middle colic artery or a common trunk between the celiac trunk and the superior mesenteric artery seem to be a developmental regression.

Two dorsal aortas are present in the early stages. Their ventral branches proceed to the primitive intestine and the yolk sac, and form the umbilical arteries [33,34,35]. Due to fusion of the two dorsal aortas, the majority of them are initially paired. In an embryo with a length of 3–5 mm (week 5, Carnegie stage 12 [36]), the most superior ventral pairs were observed at the level of the fifth to seventh cervical segments [33,34,35,37], and start to fuse to single medium stems. However, we still lack knowledge about the details of this process [33,34,35]. At the end of this stage, some of the visceral branches become larger and serve as roots for the celiac trunk and the mesenteric arteries [34,35,37]. This concentration of arteries for the gut enables the latter and the mesentery to rotate from sagittal to transverse position [32]. Originally the celiac trunk is formed by ventral branches from the sixth and seventh [35], or the seventh and eight segments [34]. After “caudal wandering”—which seems to be caused by descent of the gastrointestinal tract [33]—the celiac trunk is situated at the level of the 12th thoracic segment in an embryo of 17 mm length (week 6–7, Carnegie stage 18). The omphalomesenteric (future superior mesenteric) artery is primarily formed by four to five roots which are connected via a ventral longitudinal anastomosis [35,37]. It is already formed in stage 12 (4.5–5 mm, week 5) by roots from the level of the second to fifth thoracic segments and descends by stage 18 (17 mm, week 6–7) to its final position, which is at the first lumbar segment [33,34,37].

According to Tandler [37], the anlage of the celiac trunk is not the connection between the seventh and eighth segmental ventral branch as mentioned above, but is rather formed by the uppermost root of the omphalomesenteric artery (second thoracic segment). The roots between the last segment of the omphalomesenteric artery and that giving rise to the celiac trunk, as well as the ventral longitudinal anastomosis vanish. This results in separating the celiac trunk from the superior mesenteric artery. The same author [22] described the development of the celiacomesenteric trunk in the mole—which is the regular condition in this animal as stated above—by this mechanism. Its formation is due to the persistence of this longitudinal anastomosis, whereas the three cranial roots diminish. Thus, he concluded that the exceptional appearance of a celiacomesenteric trunk and similar variations of the celiac trunk and its branches as well as certain aberrant hepatic arteries originating from the superior mesenteric artery, or the above-mentioned anastomotic channel, are due to the same mechanism in humans [22]. This hypothesis has been accepted by several authors [5,25,27], although the occurrence of the longitudinal anastomosis in human embryos has never been confirmed by Tandler himself. Broman [33] and Pernkopf [35] also did not observe this condition and consider it an exceptional variant in human embryological development, possibly leading to a celiacomesenteric trunk, although this is not the regular mechanism of formation for the celiac trunk [33,35]. In addition, according to some authors the coexistence of an anastomotic channel and an aberrant hepatic artery from the superior mesenteric artery contradicts the theory of Tandler [8,26].

Another explanation for the development of such variants has been proposed by Broman [33], based on the development of new roots during the descent (“caudal wandering”) of the celiac trunk and the superior mesenteric artery. When the roots of the celiac trunk move to a greater extent and those of the superior mesenteric artery to a lesser extent caudally, the possibly resulting fusion may create a common trunk. A (partly) split celiac trunk may be generated during the process of its descent along the aorta if it is formed casually by two or three new roots, which may persist subsequently. In this process, the origin of the entire trunk or some of its branches may also switch to the superior mesenteric artery [33].

As Tandler’s theory [22,37] has not been confirmed yet, the genesis of the present variant cannot be explained by a persistent longitudinal anastomosis until its occurrence is verified in humans. Broman’s theory [33] could explain its occurrence, although the exact process remains unknown. In the present case we do not believe that the aberrant middle colic artery is an enlargement of a “normal” anastomosis between the celiac trunk and the superior mesenteric artery [17], or due to pre- or postnatal stenosis [8]. Indeed, it gave rise to the dorsal pancreatic artery, but had no direct connection with the stem of the superior mesenteric artery or one of its branches within the range of their sites of origin. The reason why Tandler’s theory is still proposed to explain variants of the celiac trunk [7,10,20,23,24,28,29] is probably because it is well known and has only been questioned by some authors [8,26,33,35].

### 4.2. Renal Polar Arteries

About 25–30% of the kidneys are supplied by more than one renal artery [2,4,16,38]. In such cases two to six arteries [2,3,4,15] may enter the renal parenchyma via the hilum. The arteries may originate from any level of the abdominal aorta or the common, the external, or the internal iliac arteries, the superior and inferior mesenteric arteries, branches of the celiac trunk, or the bifurcation of the aorta [2,4]. In addition to these supernumerary hilar arteries so-called polar arteries may be present [2,3,15]; upper polar arteries (21–22%) are more common than lower polar arteries (8–10%) [4,38]. Both may arise from the renal artery (15–25%) or the abdominal aorta (14–15%) [4,38]. A unique situation (less than 1%) is the simultaneous occurrence of an upper and a lower polar artery for one and the same kidney [4], as in our case on the right side. According to Adachi [16], supernumerary renal arteries are present more frequently on one side than on both sides (30.2% vs. 7.7%). In our case, the less common inferior polar artery branching off from the abdominal aorta was present for both kidneys.

Multiple renal arteries are quite common in other species. They have been reported to occur in iguanas, in snakes, saurians, crocodiles, turtles, salamanders, cartilaginous fish, and teleosts [33].

The presence of supernumerary renal arteries may be explained by the embryological development of the kidneys. Many small arteries originating from the lateral aspect of the dorsal aorta supply the mesonephros (Wolffian body). In an embryo of 5 mm length (Carnegie stage 12, week 5) they arise at the level of the second to eight [33], or first to 12th thoracic segment [37]. In stage 16 (10 mm long, week 6) these arteries originate from the level of the first to second lumbar segment and persist for the entire duration of the individual’s life, whereas the rest of them vanish between an embryo length of 16–19 mm (stage 19–20, week 7) [34].

Thus, the occurrence of the three additional renal arteries observed in our case may be explained by the persistence of these regularly vanishing arteries [33,34].

### 4.3. Corona Mortis

Usually the obturator artery arises from the internal iliac artery or one of its branches [4,15,16]. However, it may also arise from the inferior epigastric artery (11.9–29%) or the external iliac artery (0.4–2%) [3,4,15,16]. In 1–1.4% of cases it is equally formed by an anastomosis of the regular branch from the internal iliac artery and a branch of the inferior epigastric artery [4,15]. The latter is an enlargement of the regular small anastomosis of the pubic branches of these two vessels [21]. Haller [39] is said to have been the first to describe this variation [3]. A corona mortis, as in the present case, is by no means an extraordinary variation. We have observed it repeatedly in our student dissection courses. Its presence in this case is worthy of mention because of the co-occurrence with all the other variations.

In reptiles, the obturator artery is a constant branch of the internal iliac artery. In mammals, its origin is quite variable [40,41].

The obturator artery apparently does not develop before stage 22–23 (25–35 mm, week 8). A satisfactory explanation of the development of a corona mortis is still lacking. Poynter [3] has observed a single case of an obturator artery from the pubic branch of the epigastric artery forming a corona mortis in the last embryonic stage (stage 23, week 8, 32 mm). We conclude that post-embryonic processes, such as a stenosis do not form this common variation.

### 4.4. Aberrant Right Subclavian Artery and Bicarotid Trunk

A bicarotid trunk as in our case has been reported in less than 1% of individuals [4,15]. In rare cases (0.2–2.2%), the right subclavian artery arises independently distally to the origin of the left subclavian artery, directly from the upper or posterior part of the aortic arch or even from the descending aorta [2,4,15,16,42,43,44]. This aberrant artery—also called lusorial artery—courses behind the esophagus in about 80% of individuals, otherwise between the esophagus and the trachea (about 15%), and very rarely in front of the trachea [2,4,15].

Our specimen showed the rare condition of a characteristic lusorial artery with all reportedly common additional variations, namely a common stem of both common carotid arteries [44], a right non-recurrent inferior laryngeal nerve [4,16,42,43,45], a divided thoracic duct [16,44], and an S-shaped right vertebral artery [16].

The left subclavian artery may arise separately as the last branch from the aortic arch in marsupials, xenarthrans, rodents, carnivores, ungulates, prosimians, primates, and frequently gibbons. In the last two types, the common carotid arteries may arise separately (carnivores) or as a bicarotid trunk (pig, cow, horse, sheep, goat, and occasionally cat) from the brachiocephalic trunk [40,41,46]. The bicarotid trunk is a common structure among animals, which is in contrast to its prevalence in humans. A typical lusorial artery has not been reported for any species. Perhaps the latter is, as in humans, also an exceptional variant in animals and has therefore not been observed so far.

The right subclavian artery is formed like a puzzle in embryos of 10–15 mm length (stage 16–18, week 5–6), by the right fourth aortic arch, the right dorsal aortic root and the sixth (seventh) intersegmental lateral branch from the dorsal aorta. Between 14 and 18 mm length (stage 18–19, week 6–7), the two horns of the aortic sac form the later common carotid arteries. On the right side, the subclavian artery is incorporated into the proximal segment of the right horn, thus establishing the brachiocephalic trunk. In the case of a lusorial artery, the fourth aortic arch is lost and the right dorsal aorta seems to persist, giving rise to the right subclavian artery [34,36]. This origin then ascends onto the level of the left subclavian artery, and the right subclavian artery usually passes dorsally to the esophagus [36]. Probably due to lack of incorporation of the right subclavian artery, the right common carotid artery tends to shift to the left and fuse with the left common carotid artery. The coincidence of a lusorial artery and a bicarotid trunk as mentioned by Poynter [44] seems to underscore this assumption.

The recurrent laryngeal nerve courses caudally to the sixth aortic arch. Due to the disappearance of the distal portion of this arch, the nerve winds on the right side around the fourth aortic arch—the later subclavian artery. In individuals presenting with a lusorial artery due to a failure in the fourth right aortic arch, the laryngeal nerve may remain at a higher level without a recurrent course [36]. These mechanisms seem to have led to the neurovascular pattern observed in the present case.

### 4.5. Variations in the Formation of the Thyrocervical Trunk

A “regular” thyrocervical trunk [47] as described in most anatomical textbooks is only present in 30–64.5% of individuals [4,16]. In 23–45% of individuals the thyrocervical trunk may be incomplete or split [4,16]. The internal thoracic artery may also be a branch of the thyrocervical trunk in 12–23% [4,15,16,48], as in our case. In 7–15% of individuals the suprascapular artery may be a direct branch of the subclavian artery [15,16], or may even be absent in 4% of cases [3]. According to Adachi [16], in nearly all of cases presenting with a suprascapular artery originating lateral to the anterior scalene muscle, it courses below instead of above the superior transverse scapular ligament, as we observed in our specimen. We have noted this position of the suprascapular artery in some specimens during our dissection courses, but have never correlated it with its site of origin, which is definitely an interesting detail.

In mammals, the branches of the subclavian artery originate individually or in stems, namely the costocervical or omocervical trunks [40]. The formation of a thyrocervical trunk appears to be a typical human arrangement [48].

We still lack detailed knowledge about the development of the branches of the subclavian artery. The stems of the internal thoracic artery and the thyrocervical trunk have been observed in an embryo of 15.5 mm (stage 18, week 6) [34]. The different formations of the thyrocervical trunk in humans may reflect varying degrees of fusion between its individual branches [48].

### 4.6. Aberrant Branches of the External Carotid Artery

As a rule, the cervical part of the internal carotid artery does not give off any branches [16,47]. However, in rare cases one or another branch of the external carotid artery may arise from the internal carotid, e.g., the ascending pharyngeal, occipital, or lingual arteries [2].

The occipital artery is a regular dorsal branch of the external carotid artery [4,47]. Its origin from the internal carotid artery, which was present on the right side in our case, appears to be an exceptional situation; Lippert and Pabst [4] registered this phenomenon in less than 1% of cases. Adachi [16] observed only a single case of this condition. 

In about 70% of cases, the ascending pharyngeal artery is a direct (medial) branch of the external carotid artery [2,4], but may also originate from the occipital (13–23%), the common carotid (7–9%), or the internal carotid artery (6–8%)—as observed on the left side in our case—or the facial arteries (0.7–2%) [2,3,4,15,16]. 

The ascending palatine artery originates from the facial artery in 70–75% of cases, and directly from the external carotid artery in 7–23% of cases. In 5–8% it arises from the ascending pharyngeal artery [2,3,4,15,16]. It may also be a branch of the occipital or lingual arteries [2,3,16], but its origin from the internal carotid artery has not been reported earlier. Sometimes it springs off in a common trunk with the ascending pharyngeal artery (8.1%), which was noted on the left side in our case. This common stem originates from the external carotid artery in 60%, from the occipital artery in 30%, and from the facial artery in 10% of cases [16]. A common trunk of the ascending pharyngeal and palatine arteries from the internal carotid artery has not been mentioned earlier.

In summary, branches arising from the cervical portion of the internal carotid artery are an exceptional condition. In our case we even found two arteries from both internal carotid arteries.

The origin of the occipital, ascending pharyngeal, and ascending palatine arteries is very variable among vertebrates and does not show a stable pattern [40,41,49]. This may serve as an explanation for the fact that, in humans, several branches of the external carotid artery may shift to the cervical part of the internal carotid artery.

The occipital, ascending pharyngeal, and ascending palatine arteries are originally vessels of the third branchial arch [50]. The external carotid artery is a part of the dorsal aortic root, whereas the internal carotid artery is generated mainly from the third aortic arch [50,51,52].

The same derivation—namely the third aortic arch—of the internal carotid, occipital, ascending pharyngeal, and ascending palatine arteries might serve as a predisposition for the observed arterial variants.

### 4.7. Ontogenesis and Phylogenesis

As shown in Table 1, the definitive arterial pattern develops around the sixth week of gestation. Exceptionally, the obturator artery seems to develop as late as in the eighth week of gestation but knowledge about the exact time point is still lacking [3]. The critical period for the formation of all the other variations is as long as two to three weeks. Thus, it is difficult to determine exactly if the pattern has been altered in all instances at the same time, e.g., by external influence. In addition, it was not possible to ascribe all or even one of the variations to a specific species of non-human vertebrates (Table 2), which would be indicative of a mere atavism. In contrast to singular arterial variations, accumulating aberrant arteries in one individual neither reflect the regular pattern of a certain vertebrate nor a development at the same time. To the present state of knowledge, it seems that arterial variations are prone to random development.

Our efforts to clarify the formation of these variations was rendered difficult by the following facts. Despite several existing publications, we lack detailed knowledge about many processes underlying the formation of arterial patterns. We cannot confirm the validity of the information concerning comparative anatomy. Frequently we lack data as to how many individuals per species were observed in previous reports. In fact, some of these reports may have been based on single observations. Further investigations in the sectors of developmental and comparative anatomy will be needed to clarify these aspects.

### 4.8. Clinical Aspects

Arterial variations are clinically relevant, despite their different incidences. This is true for all kinds of surgery, but also for all other medical disciplines. Severing an unexpected artery may lead to severe blood loss, and in the worst case to the death of the patient. The well-known term corona mortis, representing a frequently occurring atypical origin of the obturator artery, is expressive of the potential risk of life-threatening bleeding during surgical procedures in this region [53]. Variants concerning the origin, course, and branching pattern of the celiacomesenteric arteries may lead to atypical symptoms in case of arterial occlusion. To get a valid angiography of a middle colic artery arising from the celiac trunk as in our case, cannulation of the superior mesenteric artery alone would be insufficient [9]. If the celiac trunk is split in two or more trunks, cannulation may even be difficult. The possibility of present renal polar arteries should always be kept in mind during kidney transplantation, stenting, or other endovascular procedures within the abdominal aorta. The presence of three additional renal arteries as in the present case will certainly increase the potential risk of occluding one of them during implantation of a stent. Renal arteries are end-arteries, which means that they have no anastomoses within the kidneys. Thus, occlusion of one of these arteries will result in necrosis of the supplied renal segment [4]. The replaced origin of the right subclavian artery may lead to problems during catheterizing through the right radial artery, as the catheter supposedly has to be directed in an atypical way. The associated non-recurrent inferior laryngeal nerve could be lacerated, if intraoperative neuromonitoring signals are absent due to its aberrant course [45]. Unexpected branches from the cervical part of the internal carotid artery could be severed performing endarterectomy of this artery.

## 5. Conclusions

For the first time, we report a specific appearance of a split celiac trunk, involving the origin of the middle colic and both inferior phrenic arteries: the right division formed a hepatosplenocolic trunk, giving rise to the right phrenic artery, and the left division formed a gastrophrenic trunk. Additionally, for the first time, an accumulation of replaced branches from the external carotid artery was observed. The origin of the ascending palatine artery from the internal carotid artery has hitherto not been described, neither as an individual artery, nor as part of a common trunk with the ascending pharyngeal artery as presented herein. In addition, most of the other observed arterial variations have been reported to occur rather seldom (1–2%): a bicarotid trunk, a lusorial artery, the simultaneous occurrence of an upper and a lower polar artery for one kidney, and the origin of the occipital artery from the internal carotid artery. The coexistence of all these arterial variations in the same individual is certainly unique. They seem to develop at random during embryogenesis. In sum, they do not express the regular pattern of any singular species of vertebrates.

In addition, the present study underlines the importance of appreciating anatomical variations and realizing that the human body is neither a textbook of anatomy nor an atlas. It furthermore highlights the indispensability of performing series of anatomical dissections. Surgeons, radiologists, and other health professionals must always be aware of anatomical variations during diagnostic and invasive procedures. The absence of such awareness can create major difficulties, e.g., during surgery or radiographic diagnosis.

## Figures and Tables

**Figure 1 medicina-56-00449-f001:**
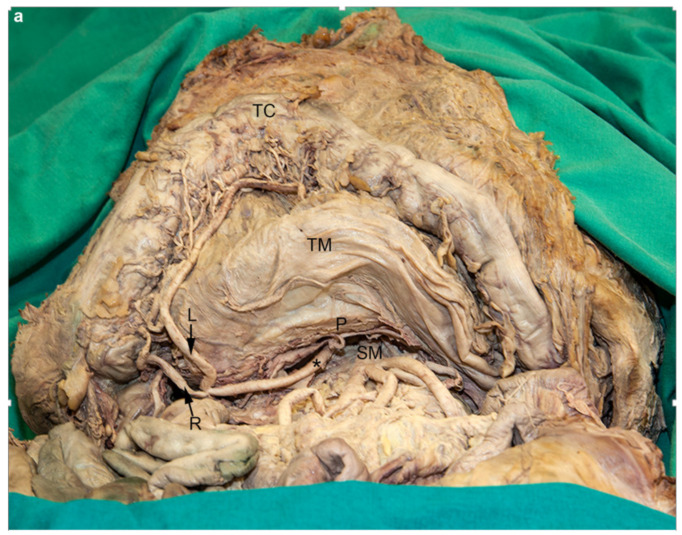
Replaced middle colic artery. (**a**) During dissection of the superior mesenteric artery (SM) we found no artery originating from its ventral aspect. The middle colic artery (asterisk) was present in the transverse mesocolon (TM), dividing into its regular branches (R, right branch; L, left branch). Proximally it disappeared into the pancreatic notch. The transverse colon (TC), together with the greater omentum were retracted cranially and the pancreas (P) was partly mobilized from its secondary retroperitoneal position. (**b**) The replaced middle colic artery (asterisk and distally between the tips of the forceps) arose together with the splenic artery (S) in a trouser-shape formation from a hepatosplenocolic trunk (arrowhead). The liver was raised to obtain a better overview. CH, common hepatic artery; TC, transverse colon; TM, transverse mesocolon.

**Figure 2 medicina-56-00449-f002:**
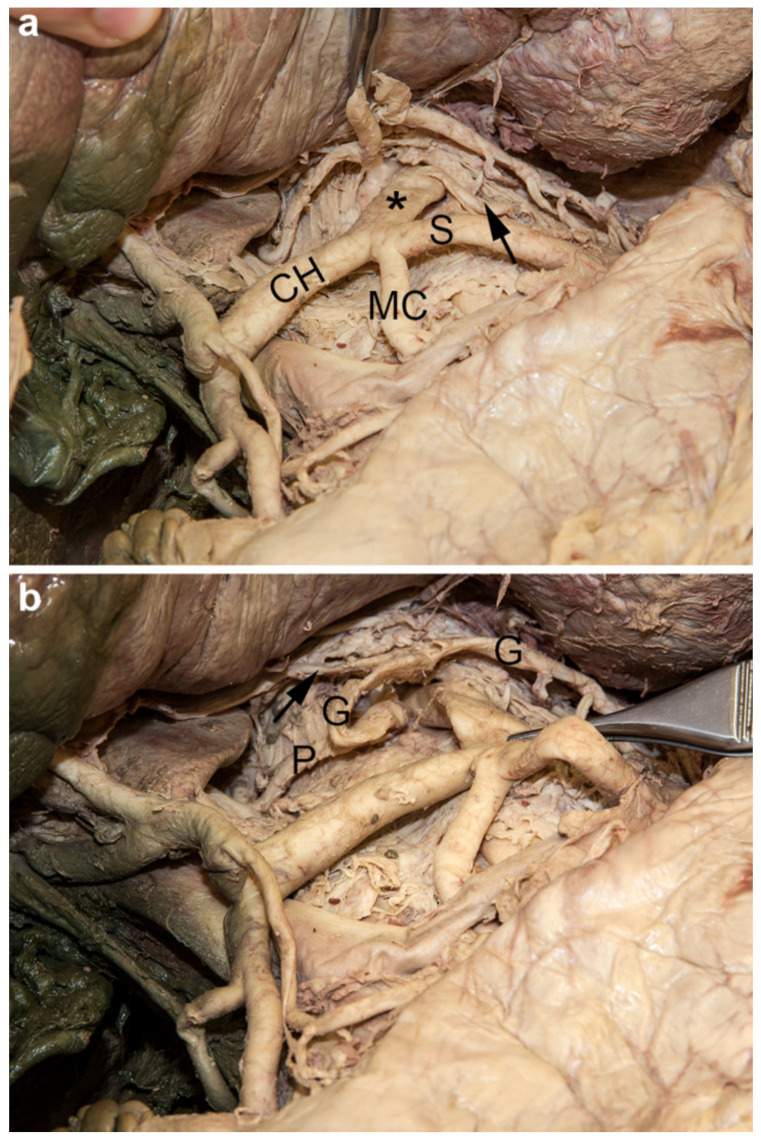
Split celiac trunk. (**a**) The celiac trunk was split into two trunks. The larger hepatosplenocolic trunk (asterisk) on the left side gave rise to the common hepatic (CH), splenic (S), replaced middle colic (MC), and left phrenic (arrow) arteries. (**b**) The smaller gastrophrenic trunk on the right side divided into the left gastric (G) and right phrenic (P) arteries. The left gastric artery gave rise to an accessory left hepatic artery (arrow).

**Figure 3 medicina-56-00449-f003:**
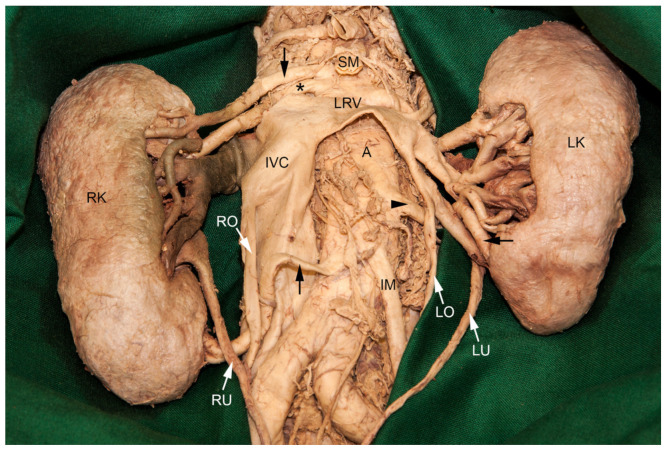
Blood supply to the kidneys. The additional left renal artery (arrowhead) gave rise to a branch for the anterior inferior segment and a lower polar artery (arrow from the right) supplying the inferior segment of the left kidney (LK). It coursed dorsally to the left ovarian vein (LO) and the lower branch of the left renal vein (LRV), and ventrally to the left ureter (LU). The right kidney (RK) received an additional artery (arrow from above) which originated cranially to the right renal artery (asterisk) from the ventral aspect of the abdominal aorta (A). It supplied the superior segment via an upper polar artery and the two anterior segments via the renal hilum. The lower polar artery (arrow from below) coursed dorsally to the right ovarian vein (RO) and the right ureter (RU), to reach the kidney parenchyma below its hilum and supply the inferior segment. Regrettably, the very small ovarian arteries did not survive the student’s dissection procedure. The inferior vena cava (IVC) was cut proximally to the termination of the left renal vein (LRV). IM, inferior mesenteric artery; SM, superior mesenteric artery.

**Figure 4 medicina-56-00449-f004:**
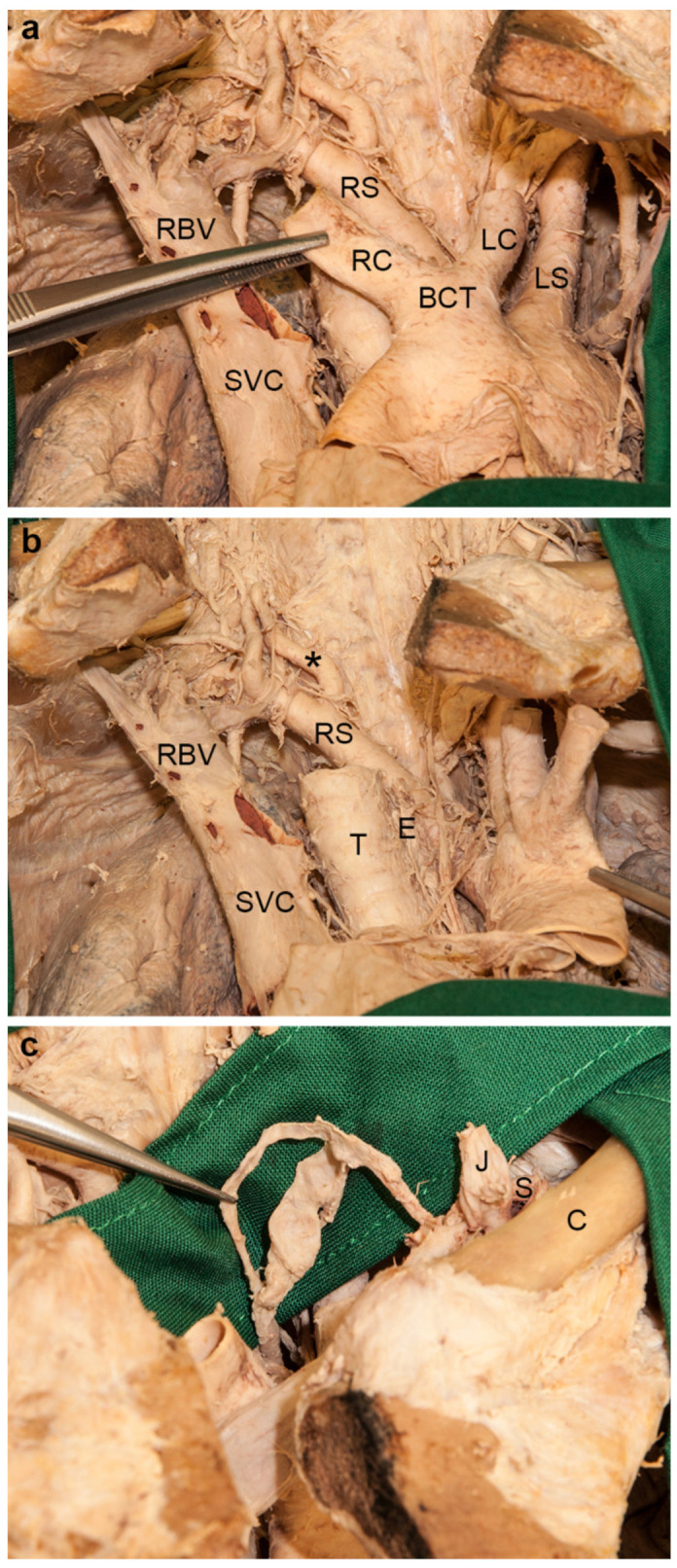
Branches of the aortic arch and termination of the thoracic duct. For a better overview, the left brachiocephalic vein was cut off at its site of confluence with the right brachiocephalic vein (RBV). (**a**) The first branch of the aortic arch was a bicarotid trunk (BCT) giving rise to the right (RC) and the left common carotid (LC) arteries. The left subclavian artery (LS) arose subsequently. The last branch was the right subclavian artery (RS). (**b**) The aortic arch was turned to the left side to show the origin and course of the aberrant right subclavian artery (RS), which coursed dorsally to the esophagus (E). The right vertebral artery (asterisk) coursed in an S-shaped manner. (**c**) At its terminal bend, the thoracic duct divided, but united again before its confluence in the left venous angle. C, left clavicle; J, left internal jugular vein (cut); S, left subclavian vein; SVC, superior vena cava; T, trachea.

**Figure 5 medicina-56-00449-f005:**
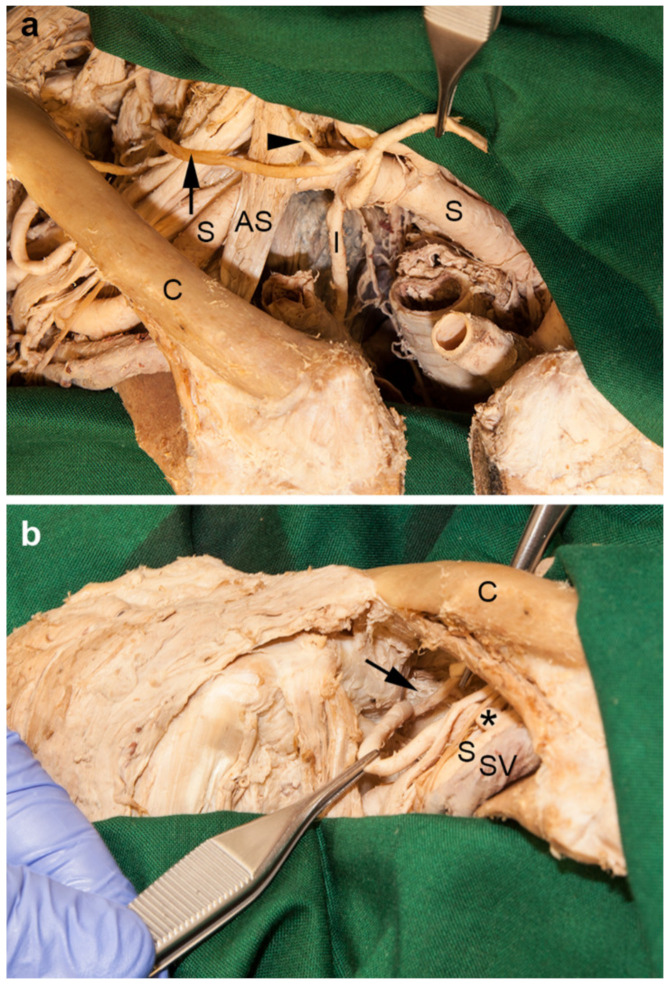
Variations of the branches of the right thyrocervical trunk. (**a**) The first branch of the right thyrocervical trunk was the internal thoracic artery (I). It then subdivided into the inferior thyroid (between the tips of the forceps) and transverse cervical arteries (arrow). The latter gave rise to the ascending cervical artery (arrowhead). (**b**) The right suprascapular artery (left forceps and asterisk) arose individually from the clavicular part of the right subclavian artery (S). It ran between the anterior divisions of the superior and middle trunks of the brachial plexus. A root of the lateral pectoral nerve crossed it ventrally. The artery reached the supraspinous fossa beneath the superior transverse scapular ligament (arrow), together with the suprascapular nerve (right forceps). AS, anterior scalene muscle; C, right clavicle; S, right subclavian artery (lusorial artery); SV, right subclavian vein.

**Figure 6 medicina-56-00449-f006:**
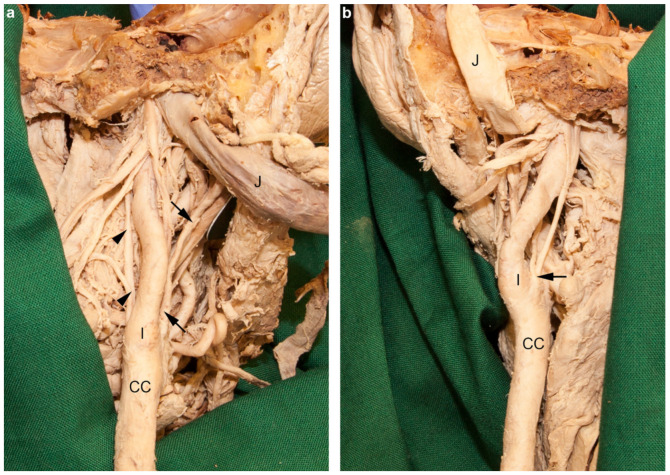
Branches of the internal carotid arteries. Parapharyngeal space from dorsal. The anterior part of the head together with the soft tissues of the neck were separated from the cervical spine. (**a**) On the right side the occipital (arrows) and ascending pharyngeal arteries (arrowheads) arose from the cervical part of the internal carotid artery (I). The internal jugular vein (J) was mobilized laterally. (**b**) On the left side the ascending pharyngeal and ascending palatine arteries arose in a common stem (arrow) from the internal carotid artery (I), close to the carotid bifurcation. The internal jugular vein (J) was turned cranially. CC, common carotid artery.

**Table 1 medicina-56-00449-t001:** Developmental timetable of arteries pertinent to the variants observed in the present case.

Arteries	Gestation Week ^1^	Length of Embryo (mm) ^1^	Carnegie Stage ^1^
Celiac trunk	5–7	3–17	12–18
Superior mesenteric artery	5–7	3–17	12–18
Renal arteries	5–7	5–19	12–20
Obturator artery ^2^	8	25–35	22–23
Branches of the aortic arch	5–7	10–18	16–19
Branches of the subclavian artery ^2^	6	15–16	18
External and internal carotid arteries	6	13–15	17–18

^1^ For references please refer to the text. ^2^ Exact knowledge about the time point of development of this entity is still lacking [3].

**Table 2 medicina-56-00449-t002:** Presence of the arterial variations as regular pattern among vertebrates.

Arterial Variation	Non-Human Vertebrates ^1^
Middle colic artery from the celiac trunk	Not observed
Supernumerary renal arteries	Common among vertebrates
Obturator artery from the external iliac	Possible among mammals
Bicarotid trunk	Saurian, quoll, manatee, African elephant, pig, cow, horse, sheep, goat, frequently cat
Lusorial artery	Not observed
Internal thoracic artery from the thyrocercival trunk	In vertebrates always directly from the subclavian artery; No thyrocervical trunk, but costocervical and/or omocervical trunks
Suprascapular artery directly from the subclavian artery	Ruminants, pigs, sometimes carnivores
Occipital artery from the internal carotid artery	Urodeles, birds, armadillo, sometimes horse; common stem in pig, reindeer, hedgehog; replaces the internal carotid artery in ruminants
Ascending pharyngeal artery from the internal carotid artery	Sometimes a common stem in the cat
Ascending palatine artery from the internal carotid artery	Not observed

^1^ For references please refer to the text.

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
