# Peer review of "A Phylogenetic and Ontogenetic Perspective of the Unique Accumulation of Arterial Variations in One Human Anatomic Specimen"

_medicina, 2020, doi:10.3390/medicina56090449_

Round 1

Reviewer 1 Report

This is a very interest manuscript about “ A phylogenetic and ontogenetic perspective of the unique accumulation of arterial variations in one human anatomic specimen”.

The dissection of the specimen is meticulous and the pictures having high quality. This is a very rare case with so many vascular variations. The manuscript is well written with appropriate introduction and discussion.

Finally there is a clear clinical message. My only objection is the manuscript is too long for a case report. My suggestion is that have to be shorten 20-25% than the present version. The manuscript can be consider after this minor revisions.

Author Response

Reviewer 1

This is a very interest manuscript about “ A phylogenetic and ontogenetic perspective of the unique accumulation of arterial variations in one human anatomic specimen”.

The dissection of the specimen is meticulous and the pictures having high quality. This is a very rare case with so many vascular variations. The manuscript is well written with appropriate introduction and discussion.

Finally there is a clear clinical message. My only objection is the manuscript is too long for a case report. My suggestion is that have to be shorten 20-25% than the present version. The manuscript can be consider after this minor revisions.

We condensed the discussion as far as possible, besides the main finding, i.e. replaced middle colic artery from a split celiac trunk. In doing so, we have omitted lines 320-325 (partly), 391-392, 400-407, 421-427, 431-432, 436-439, 441-444, 450-457, 462-474, 502-503, 505, 508-512, 517-525, 539-540, 555-566, 572-574. In addition, we added in the Conclusion, which variations have been described for the first time, and which were already known but rare (lines 620-631). We are afraid, that any further shortening will omit relevant information.

Reviewer 2 Report

Interesting report on arterial anatomical variations by Pretterklieber et al. Although it only presents 1 case, multiple anatomical variations of interest were found, making it an interesting report.

However, I have some (minor) suggestions/adjustments.

  • Please provide more detail on how the dissection was performed. Were organs resected en bloc, separately etc. Please also include the formaline % in which the body was embalmed
  • Is it possible to provide more clinical information on the patient? (in order to give an estimation about symptomatology related to the anatomical variations)
  • Please restructure the method section and carefully decide what needs to be in the methods and results section (e.g. de observations regarding the anatomical variations)

  • In all the manuscript it rather lengthly and to me a clear focus is missing. I understand that it is interesting to evaluate the phylo and ontogenetics, but please emphasize the why. Furthermore, carefully evaluate what is interesting to report to support you main message and what not. Also emphasize the novelty of the findings. Really state the most important finding and maybe group some of the other less interesting/more well described anatomical variations. 
  • I understand the focus is on arterial anatomical variations, but did you also see changes in the venous vasculature?

Author Response

Reviewer 2

Interesting report on arterial anatomical variations by Pretterklieber et al. Although it only presents 1 case, multiple anatomical variations of interest were found, making it an interesting report.

However, I have some (minor) suggestions/adjustments.

Please provide more detail on how the dissection was performed. Were organs resected en bloc, separately etc. Please also include the formaline % in which the body was embalmed

We added this information in the Material and Methods (lines 63-64, 73-83)

Is it possible to provide more clinical information on the patient? (in order to give an estimation about symptomatology related to the anatomical variations)

As we lack any clinical information besides the cause of death, we are not able to estimate any specific symptoms. The cause of death has been added (lines 62-63).

Please restructure the method section and carefully decide what needs to be in the methods and results section (e.g. de observations regarding the anatomical variations)

Thank you for this hint. We discarded listing all arterial variations in detail within this section (line 64-68), as they are reported in detail in the results section.

In all the manuscript it rather lengthly and to me a clear focus is missing. I understand that it is interesting to evaluate the phylo and ontogenetics, but please emphasize the why.

The why of the evaluation of the phylo and ontogenetics has been clearly stated as aim of this study in the final section of the Introduction (lines 55-59). We hope, this is sufficient. Please find here a copy of this text: Furthermore, we wanted to clarify, if such an arterial pattern is the regular situation within another species of vertebrates. Therefore, brief accounts of comparative anatomy are provided to explain the phylogeny of these anatomical structures. In order to determine whether these variations had been established at the same point of development probably caused by an external influence, we reviewed the embryological formation of these structures.

This is a well-accepted practice for anatomists who deal with variations and can be found in the majority of similar publications (e.g. Pretterklieber ML, Krammer EB, Mayr R. A bilateral maxillofacial trunk in man: an extraordinary anomaly of the carotid system of arteries. Acta Anat. 1991;141(3):206-11; Pretterklieber ML, Schindler A, Krammer EB. Unilateral persistence of the dorsal ophthalmic artery in man. Acta Anat. 1994;149(4):300-5; Pretterklieber ML, Krammer EB. Sphenoidal artery, ramus orbitalis persistens and pterygospinosus muscle--a unique cooccurrence of first branchial arch anomalies in man. Acta Anat. 1996;155(2):136-44; Pretterklieber B, Martschini G, Pretterklieber ML. Congenital bilateral absence of the radial artery: a very rare variation in humans - phylogenetic and ontogenetic aspects. Cells Tissues Organs. 2017;203(3):194-202). Presenting rare cases without questioning their probable genesis would not be satisfying the interested reader.

Furthermore, carefully evaluate what is interesting to report to support you main message and what not. Also emphasize the novelty of the findings. Really state the most important finding and maybe group some of the other less interesting/more well described anatomical variations.

We condensed the discussion as far as possible, besides the main finding, i.e. replaced middle colic artery from a split celiac trunk. In doing so, we have omitted lines 320-325 (partly), 391-392, 400-407, 421-427, 431-432, 436-439, 441-444, 450-457, 462-474, 502-503, 505, 508-512, 517-525, 539-540, 555-566, 572-574. In addition, we added in the Conclusion, which variations have been described for the first time, and which were already known but rare (lines 620-631). We are afraid, that any further shortening will omit relevant information.

I understand the focus is on arterial anatomical variations, but did you also see changes in the venous vasculature?

We did not observe any striking variations of the veins during the dissection.

As the course of veins is usually more variable than that of arteries (which is also stated in Gray’s Anatomy Ed. 38th, page 1574) in general only very exceptional variations are worth to be described and demonstrated to students, e.g. a persistent left inferior or superior caval vein, or a renal collar.

Reviewer 3 Report

The authors describe a highly unusual and interesting case of combined arterial variations, which they try to correlate with phylogenetic and ontogenetic aspects. However, as this attempt remains unfruitful, the main interest of the paper is that it stresses the importance of dissection training during medical education to make (future) clinicians anticipate on these kinds of variations.

My major concern with this paper is that, despite the thoroughness in describing the arterial variations the authors encountered, they seem to ignore the possibility of other concomitant variations almost completely, apart from the brief notification that "The intestines, kidneys, and thoracic organs were arranged in regular fashion."(line 65 in the document). The occurrence of arterial variations, especially when as abundant as in this case, should  a priori not be considered to be an isolated event but prompt to meticulously examine other anatomical structures, organs and systems on the presence of variations as well: were there any venous, cardiac, pulmonary, muscular or skeletal variations? (For instance, as most arterial variations occur in the medial plane along the cranio-caudal axis, I would be curious to know if there were any homeotic or meristic anomalies of the axial skeleton). This information is essential when discussion ontogenetic and phylogenetic correlations!

One minor aspect concerns the comprehensive descriptions and anatomical preparations (beautiful as they are), which would stand out much better when accompanied by schematic drawings that compare the encountered anomalies with they normally expected anatomy.

Author Response

Reviewer 3

The authors describe a highly unusual and interesting case of combined arterial variations, which they try to correlate with phylogenetic and ontogenetic aspects. However, as this attempt remains unfruitful, the main interest of the paper is that it stresses the importance of dissection training during medical education to make (future) clinicians anticipate on these kinds of variations.

Thank you for this comment. The importance of dissection training is thought to be a relevant side aspect of the conclusion, but is not the main message. Therefore, we also sum up the novelty of our findings (lines 620-631).

My major concern with this paper is that, despite the thoroughness in describing the arterial variations the authors encountered, they seem to ignore the possibility of other concomitant variations almost completely, apart from the brief notification that "The intestines, kidneys, and thoracic organs were arranged in regular fashion."(line 65 in the document).

Thank you for your comment; we have added information for the reader that no other anatomic variations have been found to be present in all regions where arterial variations were observed (lines 68-69).

The occurrence of arterial variations, especially when as abundant as in this case, should  a priori not be considered to be an isolated event but prompt to meticulously examine other anatomical structures, organs and systems on the presence of variations as well: were there any venous, cardiac, pulmonary, muscular or skeletal variations? (For instance, as most arterial variations occur in the medial plane along the cranio-caudal axis, I would be curious to know if there were any homeotic or meristic anomalies of the axial skeleton). This information is essential when discussion ontogenetic and phylogenetic correlations!

As now exactly stated in the Material and Methods (lines 68-69), no other anatomical variations have been observed. As far as the skeleton, especially the vertebral column, was accessible without macerating the whole body, we did not recognize any numeric aberrations or other structural alterations. As there is no direct connection between the arterial variations present in this case and any bony structures, we resigned to consider any further examination of the skeleton. The body has already been buried regularly after the dissection course. We hope that this is acceptable for you.

One minor aspect concerns the comprehensive descriptions and anatomical preparations (beautiful as they are), which would stand out much better when accompanied by schematic drawings that compare the encountered anomalies with they normally expected anatomy.

The regular pattern is to be found during the majority of cases in the dissection room and thus well known to the possible readership. Therefore, we will not unnecessary extend the manuscript. Please accept.

Round 2

Reviewer 3 Report

The authors substantiate in their revision that the anatomical variations at hand, unfortunately but understandably, did not come to light until an advanced state of dissection had been reached. However, it remains unclear to me if the authors  did not come across any variations in the venous circulation because they did not find any or because the veins were removed before they became aware of the arterial variations.

What remains is an exhaustive report on an accumulation of arterial variations in a single individual and the general advice for clinicians to anticipate on the occurrence of anatomical variations in daily practice (which of course has been postulated many times before).